# Clinical Value of Glycan Changes in Cerebrospinal Fluid for Evaluation of Post-Neurosurgical Bacterial Meningitis with Hemorrhagic Stroke Patients

**DOI:** 10.3390/diagnostics13020187

**Published:** 2023-01-04

**Authors:** Lei Ye, Xuefei Ji, Zijian Song, Liao Guan, Liang Zhao, Wenwen Wang, Weidong Du

**Affiliations:** 1Department of Neurosurgery, The First Affiliated Hospital of Anhui Medical University, Jixi Road 218, Hefei 230022, China; 2Department of Orthopaedics, Xuzhou Municipal First People’s Hospital, Daxue Road 269, Xuzhou 221116, China; 3Department of Pathology, Anhui Medical University, Meishan Road 81, Hefei 230032, China; 4School of Clinical Medicine, Anhui Medical University, Meishan Road 81, Hefei 230032, China

**Keywords:** bacterial meningitis, neurosurgery, hemorrhagic stroke, glycans, biochip

## Abstract

Post-neurosurgical bacterial meningitis (PNBM) is one of the severe complications in patients receiving neurosurgical procedures. Recent studies have found microbe-related glycans play important roles in adhesion, invasion, and toxicity toward innate immunological reactions. In this study, we aimed to investigate the glycomic profile and its potential diagnostic efficacy in post-neurosurgical bacterial meningitis (PNBM) patients with hemorrhagic stroke. A total of 136 cerebrospinal fluid (CSF) samples were recruited and divided into a PNBM group and a non-PNBM group based on the clinical diagnostic criteria. A lectin biochip-based method was established for the detection of glycans in CSF. The clinicopathological data and biochemical parameters in CSF from all patients were analyzed. Two models for multivariate analysis investigating glycan changes in the CSF were conducted, aiming at determining the specific expression and diagnostic efficacy of lectin-probing glycans (LPGs) for PNBM. In univariate analysis, we found that 8 out of 11 LPGs were significantly correlated with PNBM. Model 1 multivariate analysis revealed that PNA (*p* = 0.034), Jacalin (*p* = 0.034) and LTL (*p* = 0.001) were differentially expressed in the CSF of PNBM patients compared with those of non-PNBM patients. Model 2 multivariate analysis further disclosed that LTL (*p* = 0.021) and CSF glucose (*p* < 0.001) had independent diagnostic efficacies in PNBM, with areas under the curve (AUC) of 0.703 and 0.922, respectively. In summary, this study provided a new insight into the subject of CSF glycomics concerning bacterial infection in patients with hemorrhagic stroke.

## 1. Introduction

Post-neurosurgical bacterial meningitis (PNBM) is one of the severe complications among patients receiving neurosurgical treatment for the diseases of the central nervous system (CNS) [1]. Regardless of applications of advanced aseptic technology worldwide, PNBM seems inevitable, with the incidence ranging from 0.3–10% in different neurosurgical diseases [2,3]. Due to the low positive rate in bacterial culture, the etiological diagnosis for PNBM has been a great challenge. Clinical decision-making for therapeutic application of antibiotics in the disease is difficult. On the other hand, excessive neuroinflammatory reactions are commonly observed in neurological diseases, especially for those receiving invasive therapies [4]. Therefore, biochemical, and immunological components in cerebrospinal fluid (CSF) are often influenced by various confounding factors that may be resulted from both primary neurological disease and surgical procedure. Recent studies have proposed that the procalcitonin (PCT), lactate and neuron-specific enolase in CSF were promising biomarkers for diagnosis of PNBM [5,6]. However, molecular levels can be also influenced by physical and chemical changes of CSF, such as hemolysis, which is a common phenomenon during the procedure of lumbar puncture [7]. Therefore, novel biomarkers that can objectively and effectively reflect the infectious status in CSF are urgently demanded in clinical practice.

Glycosylation is a common procedure that is post-translational modification (PTM) in nature and is widely observed among mammals and microbes [8,9]. Glycan has different structures and permutations conjugating proteins, lipids or other biological components via numerous glycosyltransferases [10]. It has been reported that glycan contributes important biological functions to cell adhesion, molecular trafficking, clearance, signaling transduction and endocytosis [11]. In recent decades, increasing studies have focused on both clinical diagnostic values and molecular mechanisms of glycan in different diseases, especially in the infectious disease [12,13].

The bacterial surface is abundantly exposing glycans and glycoconjugates, which provide unique antigens against innate immunological reaction for recognition and clearance [14]. Importantly, the glycans or glycocojugates on the bacterial surface have been demonstrated to have functions in bacterial motility and adhesive ability, consequently influencing the virulence [15]. In addition, these glycans mediate the interactions of bacteria with the environment, towards both host and other bacteria, facilitating the colonization and survival [16]. In another aspect, bacteria also produces numerous glycosylated structures and polysaccharides, including capsule polysaccharide (CPS), lipopolysaccharide (LPS) and teichoic acid, which are involved in the immunological reaction and evasion [17]. However, the profiling studies of bacterial glycan were only illustrated in some pathological bacteria, such as *Neisseria gonorrhoeae* [18], *Neisseria meningitides* [16], Group B *Streptococcus* [19], *Campylobacter jejuni* [20], and *Burkholderia cepacia* [15], as well as the commensal bacteria *Lactobacillus plantarum* [21]. An evaluation of the association between glycan level and bacterial infection in CSF not only helps to understand the invasion and virulence properties but also provides a novel insight into diagnosis of infectious types and even vaccine development.

In this study, we detected different glycan levels in CSF among PNBM and non-PNBM patients receiving neurosurgery owing to hemorrhagic stroke by a well-established lectin biochip. We aimed at discovering potential biomarkers for the diagnostic values in PNBM. The results may also provide a novel concept about the profile of glycan in infectious CSF into development of glycoconjugate vaccines.

## 2. Materials and Methods

### 2.1. Patients and Sample Collection

A total of 136 patients were randomly recruited in this study. All patients were diagnosed with hemorrhagic stroke by two senior doctors and received neurosurgical treatments. CSF samples were collected via lumbar cistern drainage or lumbar puncture with aseptic technique and were stored at −80 °C. The study was conducted according to the Declaration of Helsinki and was approved by the Institutional Ethics Board of the First Affiliated Hospital of Anhui Medical University. Informed consent was obtained from all participants or the appropriate relatives.

### 2.2. Sample Treatment

Isolation of high-abundant proteins in CSF was introduced in our previous study [22]. According to the manufacturer’s protocol, high-abundant proteins, including albumin and IgG proteins, were filtrated with a Proteo Prep Immunoaffinity Albumin and IgG Depletion Kit. After that, the CSF samples were labeled with Cy3 using the protocol that was described elsewhere [22]. Redundant Cy3 was removed with a PD MiniTrap G-25 column (GE, Massachusetts, USA).

### 2.3. Profiling Glycosylation of Sera in ICH

The protocol for chemical modification on biochip was described according to our previous study [22]. Lectins, including Wheat Germ Agglutinin (WGA, L-1020), Lens Culinaris Agglutinin (LCA, L-1040), Peanut Agglutinin (PNA, L-1070), Ricinus Communis Agglutinin I (RCA-I, L-1080), Jacalin (L-1150), Vicia Villosa Lectin (VVL, L-1230), Sambucus Nigra Lectin (SNA, L-1300), Maackia Amurensis Lectin I (MAL-I, L-1310), Lotus Tetragonolobus Lectin (LTL, L-1320), Narcissus Pseudonarcissus Lectin (NPL, L-1370), and Phaseolus Vulgaris Leucoagglutinin (PVL, L-1110), were commercially obtained from Vector Laboratories Inc. (Newark, California, USA).

Each lectin was resolved in 10 mM HEPES buffer (pH 8.5) containing 0.001% BSA to the concentration of 1 mg/mL and then was immobilized on individual spot on the biochip surface, incubating at room temperature (RT) for 2 h to have robust conjugations with the surface. The biochips were then washed with a 0.01 M PBST buffer (pH 7.4) and dried with a nitrogen flow. The Cy3-labelled sera from 53 PNBM patients and 83 non-PNBM controls were individually supplied on the lectin-probed biochips and incubated in a humidity chamber and dark environment at RT for 0.5 h. After rinsed in PBST buffer twice at RT for 3 min and dried with a flow of nitrogen, the biochips were scanned with a microarray scanner (Luxscan^TM^ 10K-A microscanner, Capitalbio Co., Ltd., Beijing, China). The fluorescence intensities on the biochips were recorded.

### 2.4. Statistical Analysis

Statistical analyses were performed using SPSS 19.0 software. All continuous data were summarized as the mean ± standard deviation (Mean ± SD) or the median with interquartile range (IQR). Univariate analysis was performed using Student’s t-test or the Mann–Whitney U test. The dependent variables that were statistically significant between PNBM and non-PNBM cohort in the univariate analysis were further analyzed with multivariate analysis. Two models of multivariate analyses were conducted using lineage regression analysis. Model 1 investigated the glycan changes that were differentially expressed in the CSF of PNBM patients. Model 2 analyzed all the lectin-probing glycans (LPGs) and biochemical parameters in the CSF to discover independent risks for PNBM. B value in multivariate analysis represents the partial regression coefficient of arguments in the equation of regression. The negative value of B represents that the argument has a negative effect on the dependent variable. Binary data were analyzed using the chi-squared test. The *p* values reported in the study were two-sided and *p* < 0.05 was considered significant.

## 3. Results

### 3.1. Characteristics of PNBM Patients

This study recruited 136 patients with hemorrhagic stroke of whom 62 were diagnosed with intracerebral hemorrhage (ICH) and 74 were diagnosed with aneurysmal subarachnoid hemorrhage (aSAH). PNBM was diagnosed in 53 out of the 136 patients based on the diagnostic criteria issued by IDSA’s Clinical Practice Guidelines for Healthcare-Associated Ventriculitis and Meningitis 2017 [23] and a Chinese Expert Consensus of Diagnostic and Therapy for the Neurosurgical Central Nervous System Infections in 2021. Demographic and clinicopathological features are summarized in Table 1. Six patients had positive results for bacterial cultures, accounting for 11.3% (6/53 cases), three of whom were infected with *Stenotrophomonas maltophilia*, *Moderate thermophiles*, and *Streptococcus agalactiae*, respectively. Two patients were infected with *Acinetobacter baumannii*, and a patient was jointly infected with the bacteria of *Pseudomonas aeruginosa* and *Aeromonas caviae*. The remaining 47 patients were diagnosed with PNBM according to the biochemical characters of CSF.

### 3.2. Glycosylation Profile of CSF

We probed 11 lectins on the biochip surface to detect the differential expressions of glycans in the CSF. We found 8 out of 11 LPG levels were significantly higher in PNBM group than in non-PNBM group by the univariate analysis. They were WGA (*p* = 0.002), LCA (*p* = 0.002), PNA (*p* = 0.002), RCA-I (*p* = 0.027), Jacalin (*p* < 0.001), SNA (*p* = 0.009), MAL-I (*p* = 0.048) and LTL (*p* < 0.001). The expressions of the remaining LPGs did not reveal statistical significances between the two groups, but two revealed marginal *p* values, NPL (*p* = 0.073) and PVL (*p* = 0.086) (Figure 1). We conducted multivariate analyses of two models. In Model 1, we analyzed only the LPGs that were significantly different between PNBM and non-PNBM patients. In this model, we found that PNA (*p* = 0.034), Jacalin (*p* = 0.034) and LTL (*p* = 0.001) in CSF revealed significantly differences between the two groups. In Model 2, biochemical parameters and LPGs in the CSF that were differentially expressed between PNBM and non-PNBM groups in univariate analysis were further analyzed. The results showed that LTL (*p* = 0.021) and CSF glucose (*p* < 0.001) had statistical significances for independently distinguishing PNBM from non-PNBM (Table 2).

### 3.3. Diagnostic Values for PNBM

Based on the data from the Model 2 multivariate analysis, we found LTL and glucose in CSF were the independently hazard risks for PNBM. Therefore, we performed a receiver operator characteristic (ROC) analysis for further evaluating the diagnostic values of LTL and CSF glucose between the PNBM group and non-PNBM group. The area under the curve (AUC) was 0.703 for LTL, with sensitivity of 54.2% and specificity of 84.9% (Figure 2A), and for CSF glucose was 0.922, with sensitivity of 98.8% and specificity of 88.7% (Figure 2B).

### 3.4. Correlation Analyses for Biochemical Parameters of CSF

We performed correlation analyses between the biochemical parameters of CSF and LPGs in patients with PNBM (Figure 3A) and without PNBM (Figure 3B), respectively. We found some moderate-to-strong correlations (R > 0.7) within LPGs in both groups, indicating that there were interactions among the glycans regardless of infection status. However, there were few correlations between the CSF glucose and LPGs in both the groups, indicating that glucose might have had minor effects on glycans. Interestingly, we found some moderate correlations of LPGs with CSF protein in patients with PNBM, rather than those without PNBM (for example R = 0.511 vs. 0.163 for PNA; R = 0.435 vs. 0.184 for RCA-I; R = 0.322 vs. 0.083 for VVL R = 0.295 vs. 0.105 for SNA; R = 0.298 vs. 0.100 for MAL; R = 0.461 vs. 0.043 for NPL; R = 0.324 vs. 0.077 for PVL). This result provided evidence that the CSF glycans in the PNBM patients might be derived from the glycoproteins.

## 4. Discussion

In this study, we profiled glycan levels in CSF in patients with hemorrhagic stroke via a well-established lectin biochip. Our results indicated that PNA, Jacalin and LTL-probing glycans were differentially expressed in the CSF of PNBM patients, rather than that of non-PNBM patients. In addition, we found that LTL-probing glycans and CSF glucose might have independently diagnostic efficacies in PNBM. This study also provided a new concept about the glycomic profiles of nosocomial bacterial infection in CSF regardless of bacterial types.

Glycosylation is an important post-transcriptional modification, functioning in cell–cell adhesion, protein folding and protein trafficking. Glycan covers almost all surfaces of cells and microbes [24]. Recent studies have demonstrated that glycans or glycoconjugates might serve as biomarkers in diagnosis of multiple diseases, especially in infectious diseases. Tang et al. [25] found that N-acetyl-d-lactosamine was mainly increased on macrophages during virulent *Mycobacterium tuberculosis* infection and could serve as a potential novel diagnostic and therapeutic biomarker. Moreover, Wang et al. [26] demonstrated that O-glycosylation of CtxB-BCAL2737a could be used to detect anti-O-glycan antibodies in human serum samples from patients with *Burkholderia-associated* infections.

The composition of bacterial cell wall varies between taxonomic groups and species, depending on the cell type and the developmental stage. Bacterial surface is enriched in glycocojugates. Molecular structures of glycan on some bacterial cell walls vary from those of glycan produced by humans [24]. Previous studies demonstrated that these glycans or glycoconjugates had significant interactions with their environments, including the host and the commensal bacteria. Lectin, which could preferentially combine glycoconjugates, such as glycolipids or glycoproteins with a specific pattern as antigen-antibody interaction, has been reported to be expressed at innate immunocytes. These glycans represent an important class of pattern recognition receptors (PRRs) and have the ability of carbohydrate recognition [27]. The initial recognition and combination between lectin of innate immunocytes and glycans on bacterial surface provide early host defense. Furthermore, bacterial glycans or glycoconjugates significantly correlate with virulence. Mubaiwa et al. reported that in the processes of *Neisseria* spp. Infection, such as *N. meningitis* and *N. gonorrhoeae*, glycans involved in all stages of colonization and progression [16]. On the other hand, the bacterial glycoconjugates have also been demonstrated to be involved in immune evasion. Some *Klebsiella* serotypes averted the recognition by host lectin via altering sugars, such as mannobiose and rhamnobiose in their capsule, leading to a decreasing response of phagocytes in the respiratory tract [28].

In our study, we used two models of multiple regression analysis to investigate the differentially expressed LPGs in CSF. In Model 1, we found 3 LPGs, PNA-probing, Jacalin-probing and LTL-probing glycans, Galβ3GalNAc, Galβ3GalNAc-Ser/Thr and α-Fucose, were differentially expressed in the CSF between PNBM and non-PNBM cohorts. These results reflected the differentially expressed molecules concerning the infection, regardless of diagnostic efficacy. In previous studies, fucosylation of bacterial related proteins or lipids, which LTL was conjugated, has been widely investigated in the bacterial infections. Vimonish et al. [29] found that *Anaplasma marginale* regulated the differential expression of tick α-(1,3)-fucosyltransferases in midgut cells, indicating that the pathogen utilized core α-(1,3)-fucose of N-glycan to infect tick midgut cells. Sun et al. [30] observed a *Salmonella* neuraminidase-associated compositional shift of macrophage glycocalyx using nano-LC-chip QTOF mass cytometry technology. The shift led to an increase in fucosylation, and a subsequently impairment for both macrophage phagocytosis and galvanotaxis. However, we did find any direct evidence for the association of α-fucose or fucosylation with bacterial infections in CNS. In addition, a previous report indicated that Galβ3GalNAc, which PNA conjugated to, had an intensive interaction with cell wall-glyceraldehyde-3-phosphate dehydrogenases in *Lactobacillus reuteri* [31]. Although we failed to find the involvement of Jacalin-probing Galβ3GalNAc-Ser/Thr in bacterial infections in PNBM, our data supported the hypothesis that the glycan profiling would potentially reflect PNBM status.

The information about the infection-related glycomics in CSF is vital for development of glycoconjugate vaccines and antibody-based drugs. As early as 1923, capsular polysaccharide (CPS) of *Streptococcus pneumoniae* was found to be reactive with anti-pneumococcus sera and then it was developed as a hexavalent CPS vaccine, which received a brief license by US Food and Drug Administration [32]. So far, approximately 10 glycan-related vaccines have been licensed for use in US, offering protection strategies against numerous pathogenic strains of Gram negative bacterium [33]. Furthermore, in recent years, monoclonal antibody (mAb) has been developed as a biological tool for clinical diagnosis and treatment of infectious diseases, especially when anti-microbial resistant occurs. Several mAb drugs have been in phase I or II clinical trials, such as F598 for *N. gonorrhoeae* targeting poly-N-acetylglucosamine [34], AR-105/Aerucin for *P. aeruginosa* targeting alginate [35], and DSTA4637S for *S. aureus* targeting β-WTA of *S. aureus* [36]. Even though we are not clear about the specific infectious bacterial types in this study, our results provide a new insight into concept for PNBM. Some glycans could be applied in the development of universal vaccines toward the patients with neurosurgical treatments for prevention of bacterial infection.

In Model 2, the biochemical characters of CSF, which were related to the diagnostic criteria of meningitis, were included. The results indicated that LTL-probing glycans and glucose in CSF might be served as independently diagnostic biomarkers for PNBM. Clinical decision-making for PNBM is mainly based on the bacterial culture and biochemical factors in CSF. However, low positive rate in conventional bacterial culture and the complicated CSF environment in neurosurgical patients make the diagnosis of PNBM perplexing. We noticed that polymerase chain reaction (PCR) and next generation sequencing (NGS) can also detect the microbial species with a higher sensitivity. However, we did not apply these techniques for detection of the infectious bacteria in CSF. Because of trace abundance of bacterial load, the host DNA may disturb the PCR amplification of bacterial DNA, thus leading to a complex background. On the other hand, due to the high-sensitivity of NGS, multiple microbes can be detected, including commensal microbes and pathogenic microbes. Moreover, because of the microbial interactions, we could not distinguish which microbe contributes to the infection. CSF glucose, a monosaccharide, is routinely tested as one of the well-established biomarkers estimating CNS infections but is less served as an independent application for the diagnosis of PNBM. In this study, we found that both glucose and LTL-probing glycan in CSF would significantly be the independent biomarkers for PNBM, indicating that the glycometabolism might play a key role in the pathogenesis of PNBM. However, whether a combinational test of LTL-probing glycan and glucose in CSF could be served as a novel criterion for the independent diagnosis of PNBM should be further validated in an augmented population.

Some limitations in the study should be carefully considered. First, although 6 out of 53 patients showed positive bacterial culture, it was too few to do statistical analysis for individual microbial stains. Moreover, it was not clear what was the detailed pathogenic microbes in the remaining 47 patients negative with PNBM according to the biochemical characters of CSF, thus it is improper to make the comparison of fluorescent intensity between the culture-positive and culture-negative samples. Therefore, the results of the glycan profile in this study did not reflect relationships to a certain bacterial type but in general were the universal biomarkers for bacterial infection. Bacteria-specific glycan profiling analysis will be expected when enough CSF samples with definite information about bacterial types are recruited, and second, the LPGs in our study referred to the molecules that contained structures of specific glycans. However, we did not know about detailed information about the molecules. As indicated from the result of correlation analysis of the LPGs with biochemical characters of CSF, a combination study of proteomics and glycomics for CSF should be conducted to investigate the functional molecules in the pathogenesis of PNBM.

In summary, this study provided new information in CSF glycomics concerning bacterial infection in patients with PNBM, which would be of potential value in development of glyco-conjugated vaccines in prevention of meningitis. Additionally, we observed that the LTL-probing glycan and glucose in CSF were two independent risk factors for clinical diagnosis of PNBM.

## Figures and Tables

**Figure 1 diagnostics-13-00187-f001:**
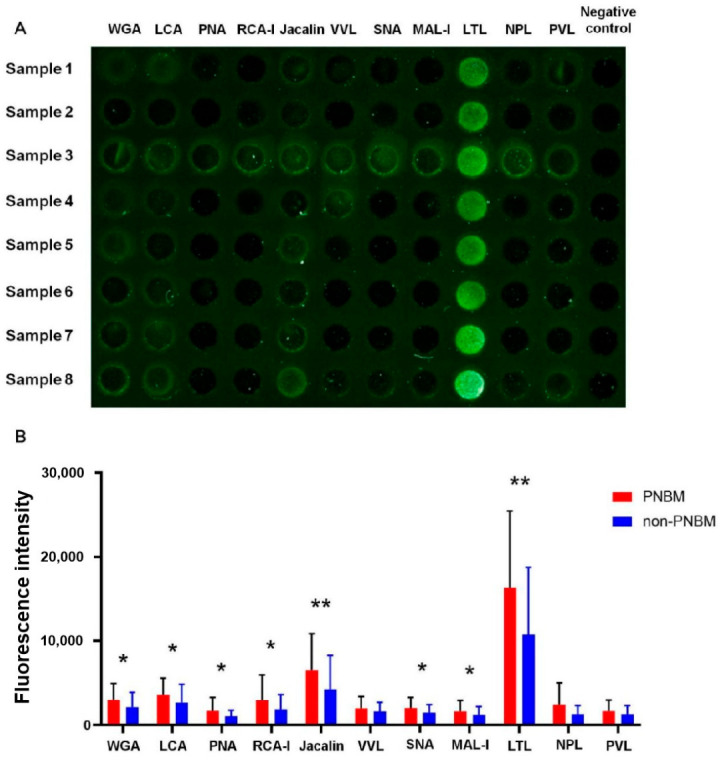
A representative fluorescence detection (**A**) and quantification comparison (**B**) of 11 kinds of lectin-probing glycans between 53 PNBM patients and 83 non-PNBM controls. Samples 1–8 belong to PNBM patients. PBS was incubated on the lectin-probed biochip instead of serum samples as negative control. (WGA: Wheat Germ Agglutinin; LCA: Lens Culinaris Agglutinin; PNA: Peanut Agglutinin; RCAI: Ricinus Communis Agglutinin I; VVL: Vicia Villosa Lectin; SNA: Sambucus Nigra Lectin; MAL-I: Maackia Amurensis Lectin I; LTL: Lotus Tetragonolobus Lectin; NPL: Narcissus Pseudonarcissus Lectin; PVL: Phaseolus Vulgaris Leucoagglutinin; PNBM: post-neurosurgical bacterial meningitis. * *p* < 0.05; ** *p* < 0.01).

**Figure 2 diagnostics-13-00187-f002:**
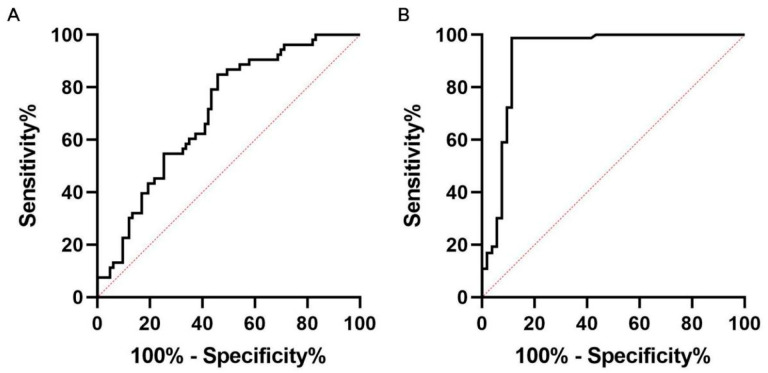
Receiver operating curve analysis for evaluating clinical values of Lotus Tetragonolobus Lectin (LTL) (**A**) and glucose (**B**) in cerebrospinal fluid in patients with post-neurosurgical bacterial meningitis.

**Figure 3 diagnostics-13-00187-f003:**
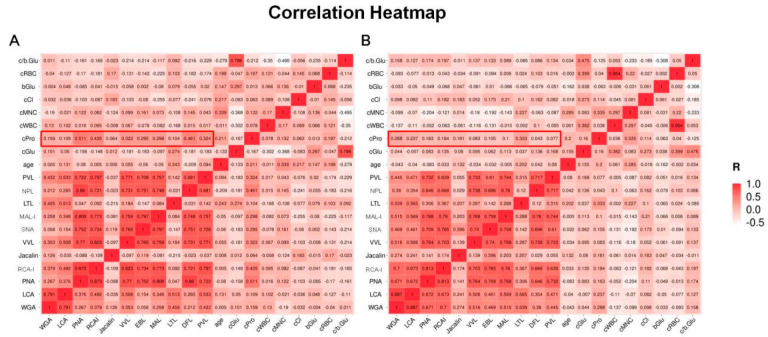
Correlation analyses between biochemical parameters and lectin-probing glycans of cerebrospinal fluid (CSF) in 53 patients with post-neurosurgical bacterial meningitis (PNBM) (**A**) and 83 patients without PNBM (**B**). (WGA: Wheat Germ Agglutinin; LCA: Lens Culinaris Agglutinin; PNA: Peanut Agglutinin; RCA-I: Ricinus Communis Agglutinin I; VVL: Vicia Villosa Lectin; SNA: Sambucus Nigra Lectin; MAL-I: Maackia Amurensis Lectin I; LTL: Lotus Tetragonolobus Lectin; NPL: Narcissus Pseudonarcissus Lectin; PVL: Phaseolus Vulgaris Leucoagglutinin; PNBM: post-neurosurgical bacterial meningitis; c/b Glu: the ratio of CSF and blood glucose; cRBC: CSF red blood cell; bGlu: blood glucose; cCl: CSF chlorine; cMNC: CSF proportion of multinuclear cell; cWBC: CSF white blood cell; cPro: CSF total protein; cGlu: CSF glucose).

**Table 1 diagnostics-13-00187-t001:** Clinicopathological characteristics of patients recruited in this study. (PNBM: post-neurosurgical bacterial meningitis; CSF: cerebrospinal fluid; cGlu: CSF glucose; bGlu: blood glucose; ICP: intracranial pressure; aSAH: aneurysmal subarachnoid hemorrhage; ICH: intracerebral hemorrhage; IQR: interquartile range).

		PNBM (*n* = 53)	Non-PNBM (*n* = 83)	*p* Value
Age (years, mean ± SD)		55.70 ± 17.09	57.65 ± 15.40	0.491
Gender				0.568
	Male	28	45	
	Female	25	38	
CSF				
	Glucose (mmol/L, IQR)	1.59 (1.26, 1.95)	3.47 (2.90, 4.33)	<0.001
	Protein (g/L, IQR)	3.00 (2.00, 4.35)	1.00 (0.63, 1.90)	<0.001
	White blood cells (×10^6^/L, IQR)	972 (345, 4768)	64 (18, 291)	<0.001
	Red blood cells (×10^6^/L, IQR)	14,000 (550, 94,500)	5000 (500, 16,000)	0.038
	Proportion of multinuclear cell (IQR)	78.2 (58.6, 88.9)	46.2 (20.0, 79.1)	<0.001
	Chlorine (mmol/L, mean ± SD)	120.30 ± 10.63	124.13 ± 8.94	0.025
	Pandy tests (negative/positive)	12/41	22/61	0.612
Blood glucose (mmol/L, IQR)		6.43 (5.43, 7.55)	7.08 (5.72, 9.22)	0.296
cGlu/bGlu ratio (IQR)		0.23 (0.16, 0.33)	0.51 (0.39, 0.64)	<0.001
ICP (mmH_2_O, IQR)		180 (115, 260)	153 (110, 210)	0.595
Primary disease				0.264
	aSAH	32	42	
	ICH	21	41	

**Table 2 diagnostics-13-00187-t002:** Multivariate analysis of lectin probing glycans and biochemical characteristics in PNBM. Model 1 of multivariate analysis included all lectin probing glycans; Model 2 of multivariate analysis was involved in combination of all the lectin-probing glycans and diagnostic characteristics for PNBM. The B value represents the partial regression coefficient of arguments in the equation of regression. (WGA: Wheat Germ Agglutinin; LCA: Lens Culinaris Agglutinin; PNA: Peanut Agglutinin; RCA-I: Ricinus Communis Agglutinin I; SNA: Sambucus Nigra Lectin; MAL-I: Maackia Amurensis Lectin I; LTL: Lotus Tetragonolobus Lectin; CSF: cerebrospinal fluid; cGlu: CSF glucose; bGlu: blood glucose).

	Model 1		Model 2	
	B	*p* Value	B	*p* Value
WGA	−0.034	0.830	−0.107	0.409
LCA	0.132	0.423	0.130	0.333
PNA	−0.356	0.034	−0.074	0.606
RCA-I	0.078	0.676	0.053	0.729
Jacalin	−0.200	0.034	−0.128	0.096
SNA	−0.129	0.431	−0.960	0.468
MAL-I	0.189	0.220	0.032	0.795
LTL	−0.324	0.001	−0.200	0.021
CSF glucose	-	-	0.394	<0.001
CSF protein	-	-	−0.108	0.198
CSF white blood cells	-	-	−0.106	0.207
CSF red blood cells	-	-	−0.036	0.661
CSF proportion of multinuclear cell	-	-	−0.097	0.199
CSF chlorine	-	-	0.100	0.146
cGlu/bGlu ratio	-	-	0.107	0.306

## Data Availability

The data that support the findings of this study are available from the corresponding author upon reasonable request.

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
