# Peer review of "Clinical Value of Glycan Changes in Cerebrospinal Fluid for Evaluation of Post-Neurosurgical Bacterial Meningitis with Hemorrhagic Stroke Patients"

_diagnostics, 2023, doi:10.3390/diagnostics13020187_

Round 1

Reviewer 1 Report

Authors present a study on the glycan change in csf in PNBM patients with hemorrhagic stroke. Analyses were conducted by a lectin panel and the results are quite interesting.

The manuscript deserves publication on diagnostics after few minor corrections.

Line 54 “Glycans, also named polysaccharides” This is not true. Please delete the sentence between commas

Line 202 “Glycosylation is an important biopolymer in the extracellular post-transcriptional modification”  please substitute with” Glycosylation is an important post-transcriptional modification”

Author Response

Reviewer 1

Authors present a study on the glycan change in csf in PNBM patients with hemorrhagic stroke. Analyses were conducted by a lectin panel and the results are quite interesting.

The manuscript deserves publication on diagnostics after few minor corrections.

Comment 1. Line 54 “Glycans, also named polysaccharides” This is not true. Please delete the sentence between commas

Answer: We greatly thank the reviewer’s appreciation. This sentence is removed.

Comment 2. Line 202 “Glycosylation is an important biopolymer in the extracellular post-transcriptional modification” please substitute with” Glycosylation is an important post-transcriptional modification”

Answer: Great thanks for the reviewer’s kind comment. We modify the sentence. Meanwhile, we have checked the whole manuscript to minimize any grammatical errors and improper expressions.

Reviewer 2 Report

In this manuscript, the authors showed the glycan changes in cerebrospinal fluid (CSF) among post-neurosurgical bacterial meningitis (PNBM) patients and non-PNBM patients using a lectin biochip-based method. The authors found three lectins, PNA, Jacalin and LTL -probing glycans, were significantly correlated with PNBM by model I multivariate analysis. Additionally, LTL and CSF glucose were independently correlated with PNBM by model II multivariate analysis. This study may provide the CSF glycan profile associated with PNBM.

Major comments:

1.     Some information provided in the method part, “2.1. Patients and sample collection” should be include in the result part, “3.1. Characteristics of PNBM patients”, such as “PNBM was diagnosed in 53 out of the 136 patients based on …” and Table 1. 

2.     In the result 3.1, the authors mentioned 6 patients had positive results for bacterial cultures. What the glycan profile of CSF did these 6 patients have? Did the authors collect the fluorescence intensity data of lectin bio-chip from these patients? Is there difference in glycan profile between these 6 patients and the rest 47 PNBM patients? Are these results can be used to connect the specific glycans with specific bacterial infection? If possible, the authors should include the data of these patients or give some discussions.

3.     In the method, “2.4. Statistical analysis”, the authors should include what data have been used for two models of multivariate analyses.

4.     In Figure 1A, Are the samples 1-8 from PNBM or non-PNBM patients? It should be defined in figure legend.  What is "negative control"? It should be noted in figure legend.

5.     In the result 3.4, the author mentioned “We performed correlation analyses between the biochemical parameters of CSF and LPGs in patients with PNBM (Figure 3A) and without PNBM (Figure 3B)”. While in the figure legend of Figure 3., it was noted as “Correlation analyses between biochemical parameters and LPGs in CSF in 136 hemorrhagic stroke patients (A) and 53 patients with PNBM (B).”  Please make it clear: which patient group are the data from in Fig3A and 3B? 

In the second paragraph of result 3.4, the authors mentioned “we found some moderate correlations of LPGs with CSF protein in patients with PNBM, rather than those with PNBM…”. Please correct this. 

6.     For some lectins, the authors used different names in the text, Figure 1, Figure3 and Table 2, such as “SNA”, “NPL” and “PHA-L” in the text, while “EBL”, “DFL” and “PVL” in figures and table. This can make the readers confused. Please keep the names consistent.

7.     In the discussion, the authors mentioned that because of low bacterial culture rate, it’s hard to connect glycan profile with certain bacterial infection. Is it possible to try PCR and NGS method? Can authors talk about this method in introduction or discussion?

Minor comments:

1.     In Table 1, some abbreviations should be defined in the table note, such as ICP, ICH, aSAH. 

2.     The authors should define what is “B” value in Table 2 note.

3.     In Figure 3, some abbreviations should be defined in the figure legend, such as cPro, cCl, cWBC, bGlu, cRBC, c/bGlu.

4.      Does any published research show the relationship of LTL-probing glycans and CSF glucose with certain bacterial infection of PNBM? If yes, can authors give some discussions?

Author Response

Reviewer 2

In this manuscript, the authors showed the glycan changes in cerebrospinal fluid (CSF) among post-neurosurgical bacterial meningitis (PNBM) patients and non-PNBM patients using a lectin biochip-based method. The authors found three lectins, PNA, Jacalin and LTL -probing glycans, were significantly correlated with PNBM by model I multivariate analysis. Additionally, LTL and CSF glucose were independently correlated with PNBM by model II multivariate analysis. This study may provide the CSF glycan profile associated with PNBM.

Major comments:

Comment 1. Some information provided in the method part, “2.1. Patients and sample collection” should be include in the result part, “3.1. Characteristics of PNBM patients”, such as “PNBM was diagnosed in 53 out of the 136 patients based on …” and Table 1.

Answer: We greatly thanks the reviewer’s appreciation for our manuscript. We agree the comment. We reorganized the section of 2.1 and 3.1 to make the manuscript much readable.

Comment 2. In the result 3.1, the authors mentioned 6 patients had positive results for bacterial cultures. What the glycan profile of CSF did these 6 patients have? Did the authors collect the fluorescence intensity data of lectin bio-chip from these patients? Is there difference in glycan profile between these 6 patients and the rest 47 PNBM patients? Are these results can be used to connect the specific glycans with specific bacterial infection? If possible, the authors should include the data of these patients or give some discussions.

Answer: We appreciate the reviewer’s nice comment. Among all the 53 patients who were diagnosed with PNBM, six patients presented positive bacterial culture. Three of the positive patients were infected with Stenotrophomonas maltophilia, Moderate thermophiles, and Streptococcus agalactiae, respectively. Two patients were infected with Acinetobacter baumannii, and a patient was jointly infected with the bacteria of Pseudomonas aeruginosa and Aeromonas caviae. However, on the one hand, due to the low abundance of bacterial load and clinical implement of antibiotics, the  rest 47 patients who were diagnosed with PNBM according to the biochemical characters of CSF had negative results for bacterial culture, and, on the other hand, because there were very limited numbers of samples of single strain infection, and we did not exactly know the detailed pathogenic microbe in the 47 patients who were diagnosed with PNBM according to the biochemical characters of CSF, we think it may be not proper to compare the fluorescent intensities of the culture-positive samples with those of culture-negative samples. We add a discussion on the study limitation.

Comment 3. In the method, “2.4. Statistical analysis”, the authors should include what data have been used for two models of multivariate analyses.

Answer: We appreciate the reviewer’s careful comment. The dependent variables that were statistically significant between PNBM and non-PNBM cohort in the univariate analysis were further analyzed with multivariate analysis. We added the expression in the section of Statistical analysis.

Comment 4. In Figure 1A, Are the samples 1-8 from PNBM or non-PNBM patients? It should be defined in figure legend. What is "negative control"? It should be noted in figure legend.

Answer: We are grateful for the reviewer’s nice comment. In order to illustrate the detection results, we provided a representative detection figure in which sample 1-8 were contained. In general, the layout of samples was random. We checked the original data and found sample 1-8 all belonged to the PNBM patients. The negative control represented that PBS was incubated on the lectin-probed biochip instead of serum samples. The negative controls were set to provide a quality control of biochips. We supplement the declaration in the figure legend.

Comment 5. In the result 3.4, the author mentioned “We performed correlation analyses between the biochemical parameters of CSF and LPGs in patients with PNBM (Figure 3A) and without PNBM (Figure 3B)”. While in the figure legend of Figure 3., it was noted as “Correlation analyses between biochemical parameters and LPGs in CSF in 136 hemorrhagic stroke patients (A) and 53 patients with PNBM (B).” Please make it clear: which patient group are the data from in Fig3A and 3B?

In the second paragraph of result 3.4, the authors mentioned “we found some moderate correlations of LPGs with CSF protein in patients with PNBM, rather than those with PNBM…”. Please correct this.

Answer: We are sorry for the mistake. We checked the original data and found that we performed the correlation analyses between the biochemical parameters of CSF and LPGs in patients with PNBM (Figure 3A) and without PNBM (Figure 3B). We modify the figure legend and corresponding context.

Comment 6. For some lectins, the authors used different names in the text, Figure 1, Figure 3 and Table 2, such as “SNA”, “NPL” and “PHA-L” in the text, while “EBL”, “DFL” and “PVL” in figures and table. This can make the readers confused. Please keep the names consistent.

Answer: We are sorry for the mistake. We checked the abbreviation and uniform the names of the three lectins as SNA, NPL and PVL in manuscript, figures, and tables.

Comment 7. In the discussion, the authors mentioned that because of low bacterial culture rate, it’s hard to connect glycan profile with certain bacterial infection. Is it possible to try PCR and NGS method? Can authors talk about this method in introduction or discussion?

Answer: We are grateful for the reviewer’s suggestive comment. Because the low abundance of bacterial load and the application of antibiotics, the positive rate in bacterial culture has been always the challenge in clinical practice. In recent decades, PCR or PCR-based NGS methods were reported to be applied in identification of microbial species. We also performed NGS for identification of microbiome in CSF samples in another study. However, we found there were two main problems in the processes of sequencing. First, because of low abundance of bacterial load, the host DNA may disturb the PCR amplification of bacterial DNA, thus leading to a complex background. Second, due to the high-sensitivity of NGS, multiple microbes can be detected, including commensal microbes and pathogenic microbes. Meanwhile, because of the microbial interaction, we could not distinguish which microbe contributes to the infection. Therefore, the PCR or NGS are still predominantly applied in the scientific researches. We supplement a discussion in the manuscript.

Minor comments:

Comment 1.In Table 1, some abbreviations should be defined in the table note, such as ICP, ICH, aSAH.

Answer: Thanks for the reviewer’s careful comment. We check all tables to supplement the full spells of abbreviation to make them much readable.

Comment 2. The authors should define what is “B” value in Table 2 note.

Answer: The B value represents the partial regression coefficient of arguments in the equation of regression. The negative value of B represents that the argument had a negative effect on the dependent variable. We add the explanation in the section of Statistical analysis and in Table 2 note.

Comment 3. In Figure 3, some abbreviations should be defined in the figure legend, such as cPro, cCl, cWBC, bGlu, cRBC, c/bGlu.

Answer: Thanks for the careful comment. The full spells are checked and supplemented in the notes of all figures.

Comment 4. Does any published research show the relationship of LTL-probing glycans and CSF glucose with certain bacterial infection of PNBM? If yes, can authors give some discussions?

Answer: We greatly appreciate the reviewer’s kind comment. LTL-probing glycans is highlighted in our study as it served as an independent diagnostic biomarker for PNBM among the patients with hemorrhagic stroke. Meanwhile, we also found that LTL-probing glycan, α-fucose, is also a hotspot among previous studies. We discuss the association between fucose- or fucosylaiton-related alterations in the infectious disease. However, so far we have not yet found any association evidence for the α-fucose or fucosylation on the pathogenesis of bacterial meningitis. We supplement a discussion in the manuscript.
